# Economic evaluation of finotonlimab plus bevacizumab as first-line therapy for advanced hepatocellular carcinoma

XueYin Xu[1]☯, Lian Tang[1]☯, XiangHua Piao[2]☯, ShaoQing Zhan[1], Yong Chen[1]*, PanFeng Feng☉[1,3,4]*

1 Department of Pharmacy, Nantong First People's Hospital, Southeast University, Nantong, Jiangsu, China, 2 Department of Pharmacy, Jiangwan Hospital of Shanghai Hongkou District, Shanghai, China, 3 Nantong Key Laboratory of Innovative Research on Rheumatology and Immunology, Nantong, Jiangsu, China, 4 Jiangsu Key Laboratory of New Drug Research and Clinical Pharmacy, Xuzhou Medical University, Xuzhou, Jiangsu, China

☯ These authors are contributed to this work equally.
* 929083891@qq.com (PF); chenyong20242024@126.com (YC)

## Abstract

### Objective

To compare the cost-effectiveness of dual-agent group (finotonlimab combined with a bevacizumab biosimilar) (SCT510) versus sorafenib as first-line treatment for advanced hepatocellular carcinoma (HCC) from the perspective of the Chinese healthcare system.

### Methods

Based on the results of a Phase III clinical trial, a three-state partitioned survival model was constructed. The primary outcomes of the model included total costs, total quality-adjusted life years (QALYs), and the incremental cost-effectiveness ratio (ICER). Cost-effectiveness analysis was employed to evaluate the economic efficiency of the dual-agent group compared to the sorafenib group as first-line treatment for advanced HCC. The model cycle length was set at 3 weeks, with a time horizon of 10 years and a discount rate of 4.5%. The willingness-to-pay (WTP) threshold was set at three times China's 2025 per capita gross domestic product (GDP) (299,400 CNY). One-way sensitivity analysis and probabilistic sensitivity analysis were conducted to assess the robustness of the results.

### Results

The ICER for the dual-agent group compared to the sorafenib group, calculated based on QALYs, was 859,053.76 CNY/QALY, which is higher than the WTP threshold (299,400 CNY). One-way sensitivity analysis indicated that parameters such as utility value in the PD state, utility value in the PFS state, the cost of finotonlimab and

**Data availability statement:** All relevant data are within the manuscript.

**Funding:** This work was supported by Jiangsu Pharmaceutical Association-Yaoyanxinsheng Fund (No.202564030), Jiangsu Pharmaceutical Association-Tianqing Fund (No. T202512), Scientific Research Project of Nantong Municipal Health and Family Planning Commission (No. MS2024038), Jiangsu Key Laboratory of New Drug Research and Clinical Pharmacy fund (No. 25KF03), Nantong Pharmaceutical Association-Yangzijiang Fund (No.ntyxky2509) and Guangzhou Zhiyi Charity Foundation fund. The funders had no role in study design, data collection and analysis, decision to publish, or preparation of the manuscript.

**Competing interests:** The authors have declared that no competing interests exist.

bevacizumab biosimilar had a significant impact on the ICER, while other parameters had minimal influence. The base-case analysis results were robust. Probabilistic sensitivity analysis showed that at a WTP threshold of 299,400 CNY, the probability of the dual-agent group being cost-effective was 0%. When the WTP threshold was approximately 842,000 CNY, the two groups had equal probability of being cost-effective. The probabilistic sensitivity analysis results were consistent with the base-case analysis.

## Conclusion

From the perspective of the Chinese healthcare system, finotonlimab combined with a bevacizumab biosimilar is not cost-effective as first-line treatment for advanced HCC.

---

## 1 Introduction

China is one of the countries with the heaviest burden of liver cancer worldwide. According to the latest statistical data, in 2022, the number of newly diagnosed liver cancer cases in China reached approximately 367,000, with more than 310,000 deaths. The incidence of liver cancer ranks fourth among all malignant tumors, while its mortality rate ranks second, especially in rural areas and among high-risk populations [1–5]. Due to the often subtle and easily overlooked early symptoms, the majority of patients are diagnosed at an advanced stage, significantly affecting treatment outcomes and survival. Currently, the overall five-year survival rate for liver cancer patients in China remains below 15%, indicating that the prevention and treatment landscape remains challenging [6–11].

Globally, sorafenib has long been regarded as the first-line standard of care for advanced hepatocellular carcinoma (HCC) [12]. However, the efficacy of this agent remains limited, with a median overall survival (mOS) generally shorter than one year [13]. Substantial evidence indicates that dual inhibition of the PD-L1/PD-1 axis and VEGF signaling pathway can synergistically enhance anti-tumor immune responses, offering a new strategic direction for the treatment of HCC [14,15]. Clinical studies have demonstrated that, in previously untreated patients with advanced hepatocellular carcinoma, toripalimab in combination with bevacizumab significantly prolongs progression-free survival and overall survival compared to sorafenib, along with a manageable safety profile [16]. Nevertheless, due to the high incidence and mortality of liver cancer, as well as factors such as heterogeneous regional regulatory approvals, restrictions in medical insurance coverage, and high treatment costs, there remains a substantial unmet clinical need [17,18]. Furthermore, owing to the high heterogeneity of hepatocellular carcinoma, not all patients benefit equally from current standard therapies. Those with severely impaired liver function, nonalcoholic fatty liver disease (NAFLD)-associated hepatocellular carcinoma, or contraindications to anti-angiogenic therapy are likely to have limited benefit [19,20]. Therefore, further optimization of treatment strategies and improvement of clinical outcomes remain important priorities in ongoing research.

Finotonlimab is a humanized IgG4 monoclonal antibody targeting PD-1, which has demonstrated anti-tumor activity in both preclinical studies and clinical trials [21].

A Phase III clinical trial conducted across 67 hospitals in China demonstrated that the combination of Finotonlimab and SCT510, a bevacizumab biosimilar, received authorization for use in China in June 2023, significantly extended median progression-free survival (7.1 months vs. 2.9 months) and median overall survival (22.1 months vs. 14.2 months) in patients, with a manageable safety profile [22]. Currently, there is a lack of pharmacoeconomic evaluation research on the Finotonlimab plus SCT510 treatment regimen in China. Therefore, from the perspective of the Chinese healthcare system, this study will apply pharmacoeconomic principles and methods to assess the cost-effectiveness of Finotonlimab in combination with SCT510 as a first-line treatment for advanced hepatocellular carcinoma, aiming to provide evidence-based support for health-care policymakers, clinical teams, and patients when selecting treatment options for advanced hepatocellular carcinoma.

## 2 Method

### 2.1 Target population and treatment regimen

Patient data and treatment regimen information were derived from a clinical study [22], which was a multicenter, open-label, Phase III randomized controlled clinical trial conducted in China involving patients with advanced hepatocellular carcinoma. The Phase III study enrolled a total of 346 patients. Patient characteristics were derived from the phase III trial [22]. Briefly, enrolled patients were aged 18 years or older with histologically confirmed advanced hepatocellular carcinoma, Child-Pugh class A liver function, and Eastern Cooperative Oncology Group performance status 0–1.

In the Phase III study, patients were randomly assigned in a 2:1 ratio to either the dual-agent group (n = 230) or the sorafenib group (n = 116). Patients in the dual-agent group received intravenous infusions of finotonlimab (200 mg) plus the bevacizumab biosimilar SCT510 (15 mg/kg) every three weeks, whereas those in the sorafenib group received oral sorafenib (400 mg) twice daily. Treatment continued until any of the following occurred: disease progression, unacceptable toxicity, initiation of new antitumor therapy, death, or loss to follow-up. According to the phase III clinical trial [22] and the Chinese Guidelines for the Diagnosis and Treatment of Primary Liver Cancer (2024 Edition) [23], the subsequent treatment costs were based on the different drugs and proportions used in the clinical trial.

As this study is entirely based on previous research [22] and publicly available data, it does not include any new research involving human participants or animals by any of the authors, and therefore does not require approval from an independent ethics committee.

### 2.2 Model structure

A three-state partitioned survival model was developed using TreeAge Pro 2022 software to simulate disease progression. The model comprised three health states: progression-free survival (PFS), progressive disease (PD), and death (D). All patients were assumed to enter the model in the PFS state. The cycle length was set to three weeks, aligned with the dosing regimen, and the time horizon was 10 years. The model structure is illustrated in Fig 1.

Key outcomes included total costs, incremental costs, quality-adjusted life years, incremental QALYs, and the incremental cost-effectiveness ratio (ICER). In accordance with the Chinese Guidelines for Pharmacoeconomic Evaluation 2025 [24], an annual discount rate of 4.5% was applied to both costs and health outcomes. The willingness-to-pay threshold was defined as three times China's per capita gross domestic product. Based on the 2025 GDP figure, the WTP threshold was set at ¥299,400 per QALY.

### 2.3 Survival analysis

Data points were extracted from the overall survival (OS) and progression-free survival (PFS) curves reported in the clinical trial using WebPlot Digitizer version 4.7. The extracted data were processed and reconstructed into individual patient

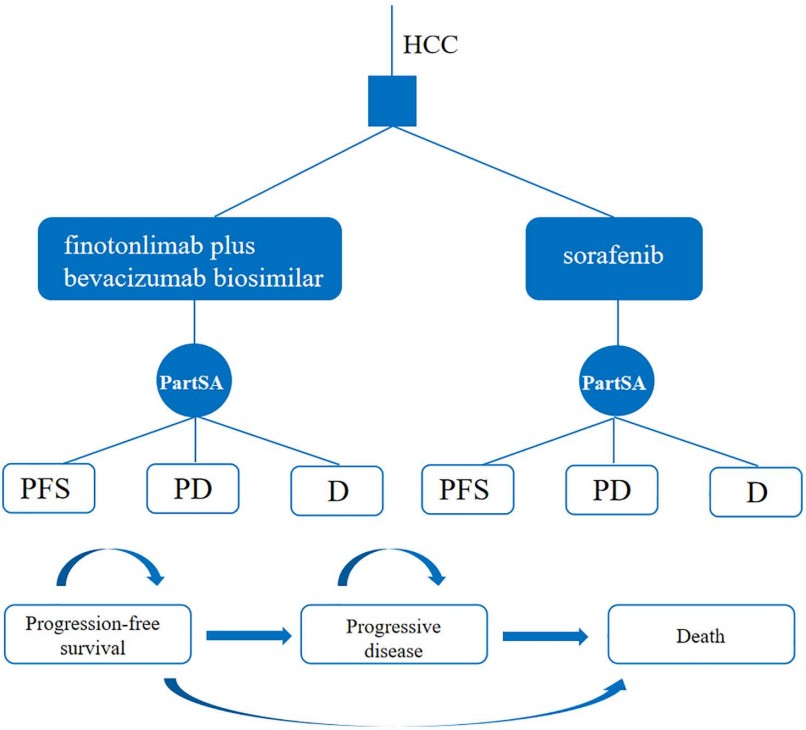

**Fig 1. Partition survival model.** PartSA, partitioned survival approach.

data using R software version 4.4.1, and subsequently used to refit the OS and PFS curves (Fig 2). Several parametric distributions—including Exponential, Gompertz, Weibull, Log-logistic, and Lognormal—were fitted to the reconstructed patient-level time-to-event data. The best-fitting distribution was selected based on the lowest Akaike Information Criterion (AIC) and Bayesian Information Criterion (BIC) values [25], supplemented by visual inspection of the fitted curves. The results of the goodness-of-fit comparison are presented in Table 1. Ultimately, the Lognormal distribution was chosen to model both PFS and OS for each treatment group. The corresponding distribution parameters are summarized in Table 2.

## 2.4 Costs and utilities

This study was conducted from the perspective of the Chinese healthcare system. Only direct medical costs were included in the cost estimation, encompassing drug costs, disease management costs [26], best supportive care expenses [27], end-of-life care costs [28], and costs associated with managing adverse events. Drug prices were based on the median 2025 tender prices across provinces in China, as published on Yaozh.com. Total treatment costs for both regimens were calculated according to the treatment duration observed in the clinical trial. For drugs dosed by body weight, a baseline weight of 59 kg was assumed [29]. Disease management costs included laboratory tests and imaging examinations. Data on adverse events (AEs) were obtained from the clinical study. Only AEs of Grade ≥3 severity with an incidence ≥5% in the original trial were incorporated. In clinical practice, Grade ≥3 AEs often lead to treatment discontinuation or switching; therefore, the model assumed a one-time cost for managing each such event, based on values derived from previously published literature [30–32]. Due to the lack of health utility data specific to the Chinese hepatocellular carcinoma population in the clinical trial, utility parameters were sourced from published studies: a utility value of 0.76 was applied for the progression-free survival state and 0.68 for the progressive disease state [33]. All model parameters and their distributions are summarized in Table 3.

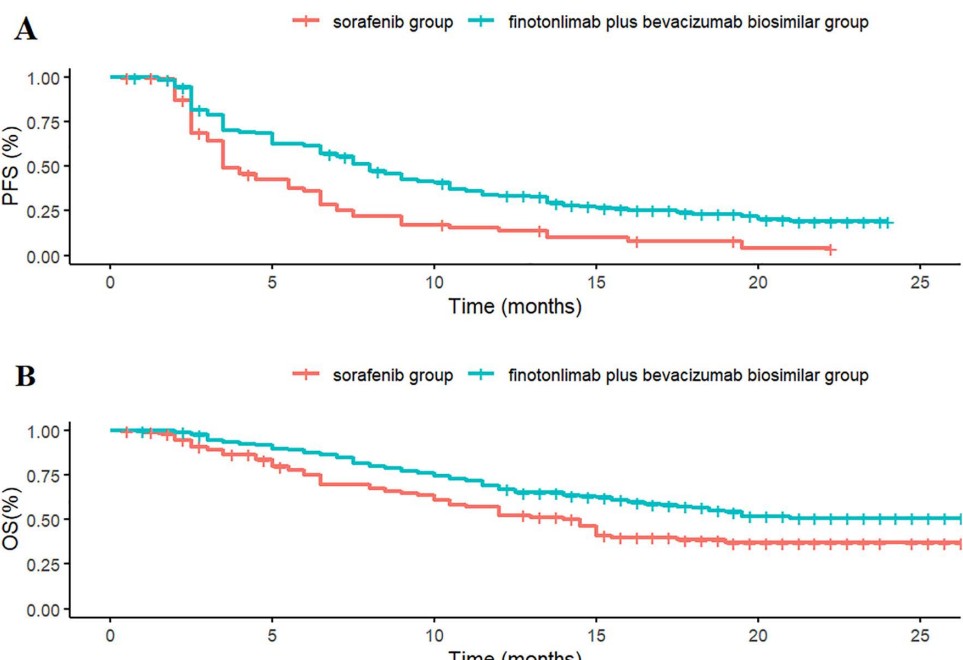

**Fig 2. Optimal curve fitting extrapolation of two treatment schemes. A.** Simulated PFS curve; **B.** Simulated OS curve.

**Table 1. AIC and BIC of survival curve in two groups.**

| Survival curve | Model criteria | Exponetial | Gompertz | Weibull | Log-logistic | Lognormal |
|---|---|---|---|---|---|---|
| Dual-agent PFS curve | AIC | 1032.664 | 1034.585 | 1027.754 | 1003.379 | 994.422 |
| | BIC | 1036.102 | 1041.462 | 1034.630 | 1010.255 | 1001.298 |
| Sorafenib PFS curve | AIC | 384.599 | 385.246 | 372.630 | 348.447 | 346.148 |
| | BIC | 387.352 | 390.753 | 378.137 | 353.955 | 351.655 |
| Dual-agent OS curve | AIC | 888.164 | 888.896 | 881.384 | 875.241 | 869.592 |
| | BIC | 891.602 | 895.772 | 888.260 | 882.117 | 876.468 |
| Sorafenib OS curve | AIC | 526.183 | 528.069 | 524.976 | 519.992 | 517.420 |
| | BIC | 528.936 | 533.576 | 530.483 | 525.499 | 522.927 |

**Table 2. Parameter distribution of survival curve in two groups.**

| Survival curve | Optimal fitting distribution | Mean | SD |
|---|---|---|---|
| Dual-agent group PFS curve | Lognormal | 2.11975 | 0.96625 |
| Sorafenib group PFS curve | Lognormal | 1.573341 | 0.703413 |
| Dual-agent group OS curve | Lognormal | 3.08907 | 1.09587 |
| Sorafenib group OS curve | Lognormal | 2.61494 | 1.09869 |

## 2.5 Scenario analysis

In this study, five parametric distributions were used to fit the PFS and OS curves, respectively, in order to evaluate their impact on the ICER. Additionally, the time horizon was set to 3, 4, 5, 6, 7, 8, 9, 10 and 15 years to further explore the

**Table 3. Model Parameters.**

| Variable | Baseline Value | Minimum | Maximum | Distribution | Reference |
|---|---|---|---|---|---|
| Cost/ CNY | | | | | |
| finotonlimab/mg | 48.800 | 39.040 | 58.560 | Gamma | Yaozhi |
| bevacizumab/mg | 10.750 | 8.600 | 12.900 | Gamma | Yaozhi |
| sorafenib/mg | 0.075 | 0.060 | 0.090 | Gamma | Yaozhi |
| Laboratory and imaging test | 605.500 | 484.400 | 726.600 | Gamma | [26] |
| Best supportive care | 6653.380 | 5322.704 | 7984.056 | Gamma | [27] |
| Terminal care | 9313.370 | 7450.696 | 11176.044 | Gamma | [28] |
| Proteinuria | 1591.000 | 1272.800 | 1909.200 | Gamma | [30] |
| Decreased platelet count | 1505.920 | 1204.740 | 1807.100 | Gamma | [31] |
| Hypertension | 256.200 | 204.960 | 307.440 | Gamma | [32] |
| Palmar-plantar erythrodysaesthesia syndrome | 27.700 | 22.160 | 33.240 | Gamma | [32] |
| Incidence rate of adverse reaction/ % | | | | | |
| Proteinuria (dual-agent group) | 6.500 | 5.200 | 7.800 | Beta | [22] |
| Decreased platelet count (dual-agent group) | 7.000 | 5.600 | 8.400 | Beta | [22] |
| Hypertension (dual-agent group) | 8.700 | 6.960 | 10.440 | Beta | [22] |
| Palmar-plantar erythrodysesthesia syndrome (sorafenib group) | 6.900 | 5.520 | 8.280 | Beta | [22] |
| Utilities | | | | | |
| PFS | 0.760 | 0.608 | 0.912 | Beta | [33] |
| PD | 0.680 | 0.544 | 0.816 | Beta | [33] |
| Others | | | | | |
| Weight | 59.000 | 47.200 | 70.800 | Normal | [29] |
| Discount rate/% | 4.500 | 0.000 | 5.000 | Beta | [24] |

influence of different simulation durations on the study results. Furthermore, changes in the ICER values were simulated under price reductions of 30%, 50% and 70% for finotonlimab and the bevacizumab biosimilar, respectively, to assess the impact of drug price negotiation on the study findings.

## 2.6 Sensitivity analysis

To verify the robustness of the base-case analysis results, both one-way sensitivity analysis and probabilistic sensitivity analysis were conducted. In the one-way sensitivity analysis, the value of a single variable was varied while keeping other parameters constant to assess its impact on the incremental cost-effectiveness ratio. The results were presented using a tornado diagram. If the upper and lower limits of a parameter were unknown, a variation of ±20% around the baseline value was assumed to define the plausible range. The outcomes of this analysis are illustrated in a tornado diagram. For the probabilistic sensitivity analysis, repeated sampling was performed based on the defined ranges and distribution types of the parameters, utilizing 1000 Monte Carlo simulations. Cost parameters were assigned a Gamma distribution, while the incidence of adverse events and utility parameters were modeled using Beta distributions. Cost-effectiveness acceptability curves and cost-effectiveness scatter plots were generated to evaluate the probability of each treatment strategy being cost-effective across a range of willingness-to-pay thresholds.

## 3 Results

### 3.1 The base case results

The base-case analysis showed that the total costs were ¥851,989.07 in the dual-agent group and ¥465,414.88 in the sorafenib group, with corresponding QALYs of 1.71 and 1.26, respectively. The dual-agent group provided an additional

0.45 QALYs at an increased cost of ¥386,574.19, resulting in an ICER of ¥859,053.76 per QALY. Since the ICER was above the pre-specified WTP threshold (Table 4), the dual-agent regimen was not considered cost-effective compared to sorafenib for the first-line treatment of advanced HCC.

### 3.2 Scenario analysis

Scenario analyses with varying time horizons demonstrated that as the simulation time extended, the ICER of the dual-agent group gradually decreased, with the magnitude of reduction diminishing over time. In all scenarios, the ICER remained above three times China's 2025 per capita GDP (Table 5). Additional analyses using alternative parametric distributions to model PFS and OS also showed that all ICER values for the dual-agent group were above the WTP threshold (Table 6).

Through price simulation, we predicted potential scenarios following the inclusion of finotonlimab and bevacizumab biosimilar into the National Reimbursement Drug List negotiations in China. We established a series of hypothetical drug price discount rates and recalculated the corresponding ICER values. The results showed that when drug prices were reduced by 30%, 50% and 70%, the corresponding ICER values were 774,887.78 CNY/QALY, 718,724.24 CNY/QALY, and 662,560.73 CNY/QALY, respectively (Table 7). Even under a 70% price reduction, the ICER remained significantly above the currently accepted willingness-to-pay threshold in China. This indicates that reducing drug prices alone is unlikely to make this regimen cost-effective under the current model assumptions.

### 3.3 One-way sensitivity analysis

The one-way sensitivity analysis indicated that parameters such as the utility value in the PD state, utility value in the PFS state, the cost of finotonlimab and bevacizumab biosimilar had the most substantial impact on the ICER. Other parameters exerted minimal influence. The results are summarized in the tornado diagram (Fig 3).

### 3.4 Probabilistic sensitivity analysis

The probabilistic sensitivity analysis was presented using a cost-effectiveness acceptability curve (CEAC, Fig 4) and a cost-effectiveness scatter plot (Fig 5). The CEAC indicated that the probability of the dual-agent group being cost-effective increased with higher WTP thresholds, while the opposite trend was observed for sorafenib. At a WTP threshold of three times China's 2025 per capita GDP (¥299,400), the probability of the dual-agent regimen being cost-effective was 0%. The two strategies had equal probability of being cost-effective at a WTP value of approximately ¥842,000. The scatter plot based on the second-order Monte Carlo simulation showed that all incremental cost-effectiveness points lay above the WTP threshold set at three times the per capita GDP, indicating that the dual-agent strategy had no economic advantage compared with sorafenib. These results were consistent with the base-case analysis.

## 4 Discussion

In China, the continuously rising costs of cancer treatment have led some patients to discontinue or even forgo therapy. Hence, selecting drugs with a cost-effectiveness advantage is particularly critical. By applying pharmacoeconomic

**Table 4. Baseline results.**

| Parameters | Dual-agent group | Sorafenib group |
|---|---|---|
| Total cost/CNY | 851989.07 | 465414.88 |
| Incremental cost/CNY | 386574.19 | – |
| Effect/QALYs | 1.71 | 1.26 |
| Incremental effect/QALYs | 0.45 | – |
| ICER, CNY/QALY | 859053.76 | – |

**Table 5. Results of scenario analyses under different simulation time horizons.**

| Simulation time horizons | Group | Total cost/ CNY | Incremental cost/CNY | Effect/ QALYs | Incremental effect/QALYs | ICER, CNY/QALY |
|---|---|---|---|---|---|---|
| 3 years | Dual-agent group | 842181.84 | 381228.25 | 1.17 | 0.22 | 1732855.68 |
| | Sorafenib group | 460953.59 | | 0.95 | | |
| 4 years | Dual-agent group | 846381.90 | 382071.26 | 1.34 | 0.27 | 1415078.74 |
| | Sorafenib group | 464310.64 | | 1.07 | | |
| 5 years | Dual-agent group | 848637.97 | 383557.41 | 1.46 | 0.32 | 1198616.91 |
| | Sorafenib group | 465080.56 | | 1.14 | | |
| 6 years | Dual-agent group | 849964.11 | 384665.40 | 1.54 | 0.36 | 1068515.00 |
| | Sorafenib group | 465298.71 | | 1.18 | | |
| 7 years | Dual-agent group | 850796.42 | 385425.40 | 1.60 | 0.39 | 988270.26 |
| | Sorafenib group | 465371.02 | | 1.21 | | |
| 8 years | Dual-agent group | 851345.52 | 385947.30 | 1.65 | 0.41 | 941334.88 |
| | Sorafenib group | 465398.22 | | 1.24 | | |
| 9 years | Dual-agent group | 851722.27 | 386312.66 | 1.68 | 0.43 | 898401.53 |
| | Sorafenib group | 465409.61 | | 1.25 | | |
| 10 years | Dual-agent group | 851989.07 | 386574.19 | 1.71 | 0.45 | 859053.76 |
| | Sorafenib group | 465414.88 | | 1.26 | | |
| 15 years | Dual-agent group | 852585.08 | 387164.21 | 1.77 | 0.48 | 806592.10 |
| | Sorafenib group | 465420.87 | | 1.29 | | |

**Table 6. Effects of different simulation distribution approaches on the ICER.**

| Distributions | Incremental cost/CNY | Incremental effect/QALYs | ICER, CNY/QALY |
|---|---|---|---|
| Exponential | 383679.12 | 0.46 | 834085.04 |
| Gompertz | 383694.96 | 0.24 | 1598729.00 |
| Weibull | 382128.06 | 0.18 | 2122933.67 |
| Log-logistic | 365598.30 | 0.38 | 962100.79 |
| Lognormal | 386574.19 | 0.45 | 859053.76 |

**Table 7. ICER values at different discount rates for drug prices.**

| Price discount rate | Group | Total cost/ CNY | Incremental cost/CNY | Effect/ QALYs | Incremental effect/QALYs | ICER, CNY/QALY |
|---|---|---|---|---|---|---|
| 70% | Dual-agent group | 814078.70 | 348699.50 | 1.71 | 0.45 | 774887.78 |
| 50% | Dual-agent group | 788805.11 | 323425.91 | 1.71 | 0.45 | 718724.24 |
| 30% | Dual-agent group | 763531.53 | 298152.33 | 1.71 | 0.45 | 662560.73 |
| | Sorafenib group | 465379.20 | | 1.26 | | |

evaluation methods to compare the economic efficiency of different treatment options, we can not only provide a reference for rational drug use in clinical practice but also help enhance the efficiency of healthcare insurance fund allocation [34]. Based on the NCT04560894 trial and from the perspective of the Chinese healthcare system, this study evaluated the cost-effectiveness of finotonlimab combined with a bevacizumab biosimilar (dual-agent group) versus sorafenib as first-line treatment for advanced hepatocellular carcinoma (HCC). The results showed an incremental cost-effectiveness

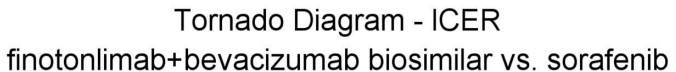

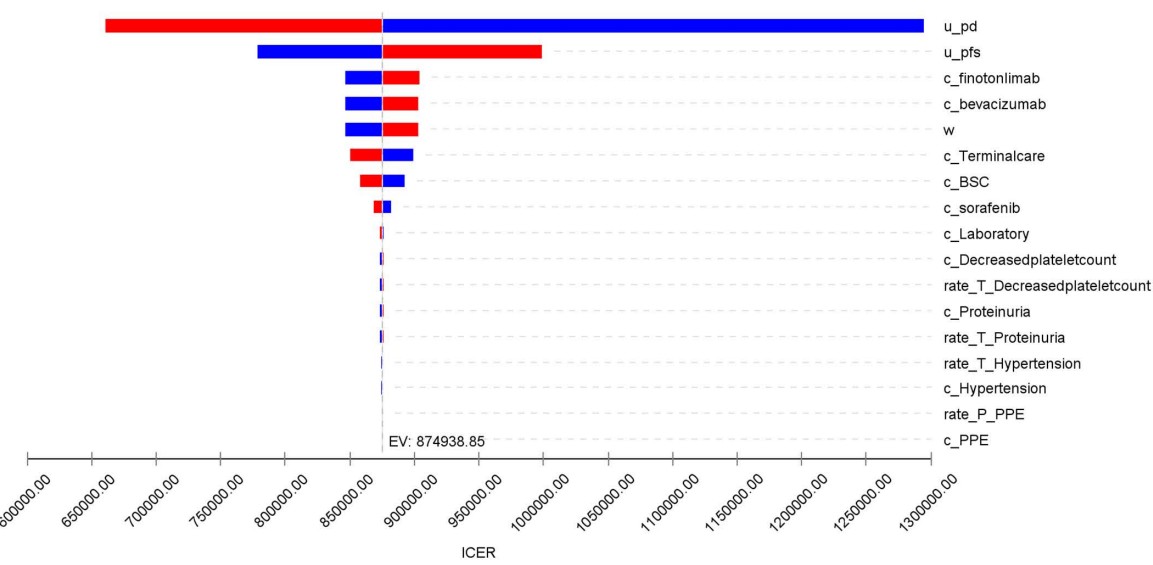

**Fig 3. Tornado diagram for one-way sensitivity analysis.**

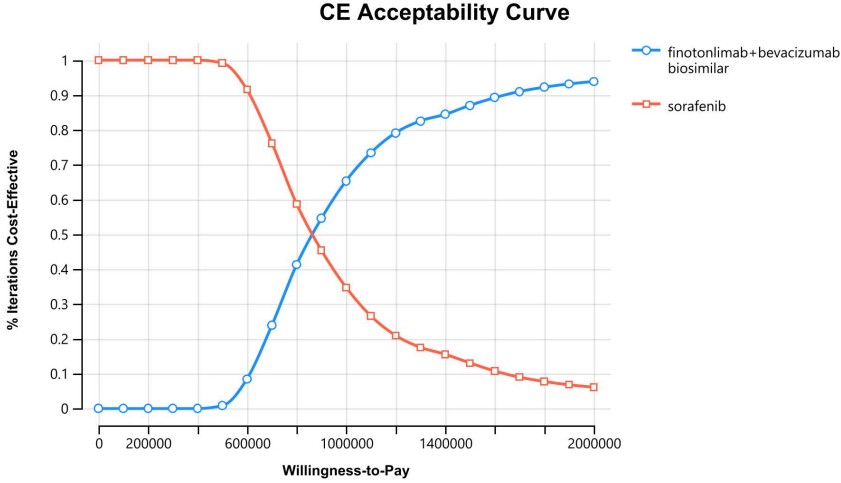

**Fig 4. Acceptability curves.**

ratio (ICER) of ¥859,053.76 per quality-adjusted life year (QALY), which was above the pre-defined willingness-to-pay (WTP) threshold. These findings suggest that the combination of finotonlimab and a bevacizumab biosimilar is not a cost-effective first-line treatment for advanced HCC.

One-way sensitivity analysis indicated that parameters such as the utility value in the PD state, utility value in the PFS state, the cost of finotonlimab and bevacizumab biosimilar had considerable impact on the results. Probabilistic sensitivity analysis demonstrated that at a WTP threshold of ¥299,400, the probability of the dual-agent regimen being cost-effective

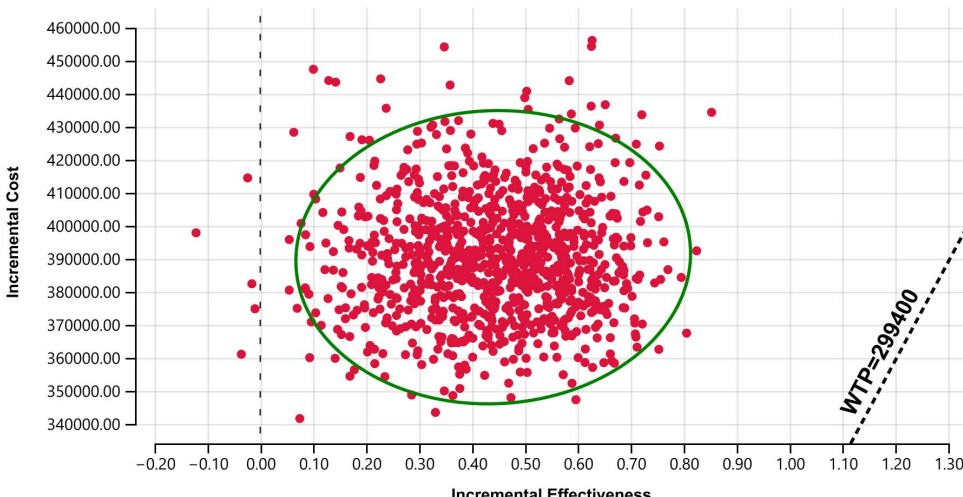

**Fig 5. Cost-effective scatter plot.** Results of Monte Carlo probabilistic sensitivity analysis showing incremental cost-effective of finotonlimab plus bevacizumab biosimilar versus sorafenib.

was 0%. When the WTP was approximately ¥842,000, both treatment strategies had equal probability of being cost-effective. The results of the probabilistic sensitivity analysis were consistent with the base-case analysis.

The results of this study indicated that although the combination of finotonlimab and a bevacizumab biosimilar extended overall survival compared with sorafenib, its incremental cost-effectiveness ratio was far above the current willingness-to-pay threshold in China, suggesting that this regimen was not cost-effective under current pricing. The findings of this study differed somewhat from existing economic evaluations supporting the cost-effectiveness of immuno-combination therapies for advanced hepatocellular carcinoma. For example, the atezolizumab plus bevacizumab regimen based on the IMbrave150 trial has become an international standard therapy, and subsequent economic evaluations conducted in China have generally shown that this regimen was cost-effective compared with sorafenib, with incremental cost-effectiveness ratios typically falling within acceptable thresholds [35]. Similarly, the sintilimab plus bevacizumab regimen based on the ORIENT-32 study was also demonstrated to be a cost-effective option in the Chinese context [36–38]. The higher ICER observed in this study was primarily attributable to the unclear pricing strategy and reimbursement status of finotonlimab as a novel PD-1 inhibitor, combined with uncertainties in the extrapolation of survival benefits, which prevented the regimen from demonstrating cost-effectiveness. This difference suggested that finotonlimab was not yet a preferred regimen in clinical practice, and further optimisation of pricing strategies or reduction of out-of-pocket costs through reimbursement negotiations was needed. This study provided preliminary evidence on the economic value of the finotonlimab combination regimen as first-line therapy for advanced hepatocellular carcinoma and served as a reference for future reimbursement decisions.

Consistent with other studies utilizing partitioned survival models, this study simulated the natural history of hepatocellular carcinoma and evaluated the cost-effectiveness of finotonlimab plus a bevacizumab biosimilar (dual-agent group) versus sorafenib as first-line therapy for advanced HCC within constrained healthcare resources. Nevertheless, several limitations should be acknowledged. First, patient data regarding progression-free survival and overall survival were derived from published clinical trials, which may introduce bias compared to real-world data. As this analysis is based on randomized controlled trials with strict inclusion/exclusion criteria and high patient adherence, the findings may not be fully generalizable to all patients encountered in routine clinical practice. Second, this study only considered direct medical

costs, while direct non-medical costs and indirect costs were excluded. This may lead to an underestimation of the total actual treatment cost per patient. The costs associated with managing adverse events were sourced from published literature rather than real-world data, which might not adequately reflect the actual medical and economic context in China. Third, only severe adverse reactions (≥ Grade 3) with an incidence rate difference of ≥3% between the two treatment groups were included in the analysis. Not all AEs were considered, and the associated management costs may differ from actual clinical practice. However, one-way sensitivity analysis demonstrated that the cost of AE management had minimal impact on the overall results. Additionally, owing to the absence of disutility values for relevant adverse events, this study did not incorporate them into the analysis. Further investigation into their impact is warranted in future studies when reliable data become available. Fourth, compared with the Markov model, the partitioned survival model simplifies the disease course into three mutually exclusive health states, which makes it difficult to capture clinical realities such as subsequent treatments, disease recurrence, or complex transitions between states. This may affect the model's ability to accurately reflect real-world clinical practice during the extrapolation period. Finally, in this study, five commonly used parametric distributions (Exponential, Gompertz, Weibull, Log-logistic, and Lognormal) were adopted for survival extrapolation, whereas seven-parameter distributions such as Gamma and Generalised Gamma were not included. Consequently, this may result in inadequate characterization of uncertainty in the tails of the survival curves.

Despite these limitations, this study thoroughly addressed uncertainties through extensive sensitivity analyses, which confirmed the robustness of the base-case findings. Therefore, the results still provide valuable references for clinical decision-making and health insurance reimbursement negotiations.

## 5  Conclusion

In conclusion, from the perspective of the Chinese healthcare system, and using a willingness-to-pay threshold of three times the Chinese GDP per capita, the finotonlimab plus bevacizumab biosimilar regimen is not a cost-effective first-line treatment option for patients with advanced hepatocellular carcinoma.

## Author contributions

**Conceptualization:** Panfeng Feng.

**Data curation:** XueYin Xu, Lian Tang, XiangHua Piao, ShaoQing Zhan, Yong Chen.

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
