## [Decision Letter · Decision Letter 0]

14 Mar 2026

PONE-D-25-65811

Economic Evaluation of Finotonlimab plus bevacizumab as First-line Therapy for advanced Hepatocellular Carcinoma

PLOS One

Dear Dr. Feng,

Thank you for submitting your manuscript to PLOS ONE. After careful consideration, we feel that it has merit but does not fully meet PLOS ONE’s publication criteria as it currently stands. Therefore, we invite you to submit a revised version of the manuscript that addresses the points raised during the review process.

We look forward to receiving your revised manuscript.

Kind regards,

Jincheng Wang

Academic Editor

PLOS One

Journal Requirements:

“Ethics approval

As this study is entirely based on previous research [8] and publicly available data, it does not include any new research involving human participants or animals by any of the authors, and therefore does not require approval from an independent ethics committee.”

4. Please note that funding information should not appear in any section or other areas of your manuscript. We will only publish funding information present in the Funding Statement section of the online submission form. Please remove any funding-related text from the manuscript.

5. We note that your Data Availability Statement is currently as follows: “All relevant data are within the manuscript and its Supporting Information files”

Additional Editor Comments:

This study presents a cost-effectiveness analysis of finotonlimab plus bevacizumab biosimilar versus sorafenib for advanced HCC using a partitioned survival model, with a well-below-threshold ICER of ¥50,492.98/QALY. However, several issues require revision. The exclusive use of lognormal distribution for all survival curves needs biological justification, particularly given its non-monotonic hazard implication for HCC. Extrapolation to 10 years is questionable given the poor prognosis of advanced HCC, and the authors should report the proportion of patients surviving beyond 5 years under each distribution. The utility values derived from a 2016 non-Chinese study are insufficiently justified given their demonstrated sensitivity impact, and alternative sources should be explored. Most critically, sorafenib is no longer the standard comparator in China — the absence of comparison against atezolizumab plus bevacizumab or sintilimab plus bevacizumab substantially limits clinical relevance. Minor issues include inconsistent AE inclusion criteria between the Methods and Discussion sections.

Reviewer's Responses to Questions

**Comments to the Author**

1. Is the manuscript technically sound, and do the data support the conclusions?

Reviewer #1: Partly

Reviewer #2: Yes

2. Has the statistical analysis been performed appropriately and rigorously? 

Reviewer #1: No

Reviewer #2: Yes

3. Have the authors made all data underlying the findings in their manuscript fully available?

Reviewer #1: Yes

Reviewer #2: Yes

4. Is the manuscript presented in an intelligible fashion and written in standard English?

Reviewer #1: Yes

Reviewer #2: Yes

5. Review Comments to the Author

Reviewer #1: Thank you for your assigning me to review the submitted manuscript. This is not a methodologically rigorous economic evaluation, with significant limitations in data fitting, subsequent treatment costs, and disutility values. Therefore, the author should reconstruct the model and update the results.

Major comments:

1. Finotonlimab combined with a bevacizumab biosimilar yields superior efficacy at a higher cost, so why does the Log-logistic distribution yield a dominant result? This is an absolutely erroneous outcome. We seriously doubt the accuracy of the overall study.

2. “Furthermore, owing to the high heterogeneity of hepatocellular carcinoma, not all patients benefit equally from current standard therapies.” Which groups of people? Please support your statement with relevant literature and data.

3. Please provide CHEERS 2020 in the supplementary materials.

4. Removing the inclusion and exclusion criteria of clinical trials, as these are unrelated to the study methodology. A brief description of patient characteristics is sufficient.

5. To our knowledge, the subsequent treatment regimen recommended in the CSCO 2024 guidelines does not best supportive care. Please calculate the subsequent treatment costs based on the different drugs (Immunotherapy, Targeted therapy, Chemotherapy) and proportions used in clinical trials (Supplementary Table 4).

6. Why choose the partitioned survival model rather than the Markov model? What are the differences between them?

7. According to the Economic Evaluation Guidelines 2025, the discount rate is 4.5%, with a range of 0-5%.

8. Please update the GDP per capita to 2025 (￥99800).

9. Typically, seven-parameter survival models are employed for data fitting. Please add Gamma and Generalised Gamma, and present the AIC and BIC values for different parameter survival models in the supplementary materials.

10. What does “Laboratory and imaging tests” include? Please specify the specific tests and costs (Blood tests, CT scans, bone scans, etc.).

11. How are the costs of adverse events incorporated into the model?

12. Why are the utility values derived from Reference 22 rather than published real-world studies? Why were disutility values associated with adverse events not considered? A rigorous economic evaluation should incorporate disutility values.

13. What is the difference between 1000 and 10,000 times second-order Monte Carlo simulations? Why choose 1000 times?

14. Please reconstruct the model, as the current results are absolutely incorrect.

15. The current discussion does not meet the standards outlined in the economic evaluation guidelines(CHEERS 2020). The author should explain why the novel treatment regimen is or is not cost-effective. The author should briefly review existing economic evaluations related to advanced hepatocellular carcinoma, identifying the optimal/preffered treatment regimen in available economic evaluations and comparing it with the findings of this study.

Reviewer #2: This manuscript presents a well-conducted pharmacoeconomic analysis suggesting finotonlimab plus bevacizumab biosimilar is cost-effective versus sorafenib for advanced HCC in China. While the methodological framework is generally appropriate, several critical issues must be addressed to ensure robustness and transparency. Below are the major and minor concerns:

1、Drug Cost Assumptions Need Validation：The model assumes full drug costs without accounting for possible charity drug programs during the trial (2022–2025), as both Finotonlimab and bevacizumab biosimilar were unapproved in China at the time. Maybe sensitivity analysis are needed, try to explore reduced pricing scenarios if possible (e.g., discount rates negotiated post-approval).

2、Incomplete Adverse Event (AE) Cost Capture：Serious AEs (≥Grade 3) reported in the Phase III trial (e.g., 27 cases of immune-related AEs and even fatal events) are not incorporated into Table 3 or model inputs. AE-associated costs (e.g., hospitalization, treatment discontinuation) may significantly impact cost-effectiveness.

3、Patient Flow and Discontinuations：The Materials & Methods section lacks clarity on:

Sample size: Exact patient numbers randomized to each arm Treatment discontinuations: Rates due to AEs/progression, which affect drug utilization and cost.

4、Some figures (particularly Figure 2) currently suffer from poor readability due to:small font sizes for axis labels and legends

6. PLOS authors have the option to publish the peer review history of their article (what does this mean?). If published, this will include your full peer review and any attached files.

Reviewer #1: No

Reviewer #2: No

---

## [Author Response · Author response to Decision Letter 1]

9 Apr 2026

We have uploaded the review comments as an attachment, named "response to comments".

---

## [Decision Letter · Decision Letter 1]

21 Apr 2026

PONE-D-25-65811R1Economic Evaluation of Finotonlimab plus bevacizumab as First-line Therapy for advanced Hepatocellular CarcinomaPLOS One

Dear Dr. Feng,

Thank you for submitting your manuscript to PLOS ONE. After careful consideration, we feel that it has merit but does not fully meet PLOS ONE’s publication criteria as it currently stands. Therefore, we invite you to submit a revised version of the manuscript that addresses the points raised during the review process.

**ACADEMIC EDITOR:** After revisions, this paper has been improved. However, there is still a problem commented by a reviewer: It is recommended that three times the per capita GDP be employed as the willingness-to-pay threshold, i.e. 99,800 × 3 = 299,400. Please address.

We look forward to receiving your revised manuscript.

Kind regards,

Jincheng Wang

Academic Editor

PLOS One

Journal Requirements:

Additional Editor Comments:

After revisions, this paper has been improved. However, there is still a problem commented by a reviewer: It is recommended that three times the per capita GDP be employed as the willingness-to-pay threshold, i.e. 99,800 × 3 = 299,400. Please address.

Reviewers' comments:

Reviewer's Responses to Questions

**Comments to the Author**

1. If the authors have adequately addressed your comments raised in a previous round of review and you feel that this manuscript is now acceptable for publication, you may indicate that here to bypass the “Comments to the Author” section, enter your conflict of interest statement in the “Confidential to Editor” section, and submit your "Accept" recommendation.

Reviewer #1: (No Response)

Reviewer #2: All comments have been addressed

2. Is the manuscript technically sound, and do the data support the conclusions?

Reviewer #1: Partly

Reviewer #2: Yes

3. Has the statistical analysis been performed appropriately and rigorously? 

Reviewer #1: No

Reviewer #2: Yes

4. Have the authors made all data underlying the findings in their manuscript fully available?

Reviewer #1: Yes

Reviewer #2: Yes

5. Is the manuscript presented in an intelligible fashion and written in standard English?

Reviewer #1: Yes

Reviewer #2: Yes

6. Review Comments to the Author

Reviewer #1: This study still contains obvious errors; please treat academic research with due seriousness. It is recommended that three times the per capita GDP be employed as the willingness-to-pay threshold, i.e. 99,800 × 3 = 299,400. What was the author’s intention in writing ‘three times per capita GDP’ as ‘two times per capita GDP’ (199,600)? (At a WTP threshold of three times China’s 2025 per capita GDP (¥199,600), the probability of the dual-agent regimen being cost-effective was 0%.)

Reviewer #2: (No Response)

7. PLOS authors have the option to publish the peer review history of their article (what does this mean?). If published, this will include your full peer review and any attached files.

Reviewer #1: No

Reviewer #2: No

---

## [Author Response · Author response to Decision Letter 2]

21 Apr 2026

Reviewer #1: This study still contains obvious errors; please treat academic research with due seriousness. It is recommended that three times the per capita GDP be employed as the willingness-to-pay threshold, i.e. 99,800 × 3 = 299,400. What was the author’s intention in writing ‘three times per capita GDP’ as ‘two times per capita GDP’ (199,600)? (At a WTP threshold of three times China’s 2025 per capita GDP (¥199,600), the probability of the dual-agent regimen being cost-effective was 0%.)

Response: Thank you for pointing out this critical issue. Based on your suggestion, we have revised the threshold to "three times the per capita GDP" throughout the manuscript, with the corresponding willingness-to-pay threshold set to 299,400 CNY. The probability of the two-drug regimen being cost-effective at this threshold has been recalculated. The revised results show that "when using three times the per capita GDP of China (299,400 CNY) as the willingness-to-pay threshold, the probability of the two-drug regimen being cost-effective is 0%."

---

## [Editor Report · Decision Letter 2]

23 Apr 2026

Economic Evaluation of Finotonlimab plus bevacizumab as First-line Therapy for advanced Hepatocellular Carcinoma

PONE-D-25-65811R2

Dear Dr. Feng,

We’re pleased to inform you that your manuscript has been judged scientifically suitable for publication and will be formally accepted for publication once it meets all outstanding technical requirements.

Kind regards,

Jincheng Wang

Academic Editor

PLOS One

Additional Editor Comments (optional):

Authors have addressed all comments, and I think this paper can be accepted for publication.
---

## [Editor Report · Acceptance letter]

PONE-D-25-65811R2

PLOS One

Dear Dr. Feng,

I'm pleased to inform you that your manuscript has been deemed suitable for publication in PLOS One. Congratulations! Your manuscript is now being handed over to our production team.

Kind regards,

on behalf of

Dr. Jincheng Wang

Academic Editor

PLOS One